# Breaking the Dilemma of Medical Image-to-image Translation

**Lingke Kong**[*]
Manteia Tech
konglingke@manteiatech.com

**Chenyu Lian**[*]
Xiamen University
cylian@stu.xmu.edu.cn

**Detian Huang**
Huaqiao University
huangdetian@hqu.edu.cn

**Zhenjiang Li**
Shandong University
zhenjli1987@163.com

**Yanle Hu**[†]
Mayo Clinic Arizona
Hu.Yanle@mayo.edu

**Qichao Zhou**[†]
Manteia Tech
zhouqc@manteiatech.com

## Abstract

Supervised Pix2Pix and unsupervised Cycle-consistency are two modes that dominate the field of medical image-to-image translation. However, neither modes are ideal. The Pix2Pix mode has excellent performance. But it requires paired and well pixel-wise aligned images, which may not always be achievable due to respiratory motion or anatomy change between times that paired images are acquired. The Cycle-consistency mode is less stringent with training data and works well on unpaired or misaligned images. But its performance may not be optimal. In order to break the dilemma of the existing modes, we propose a new unsupervised mode called RegGAN for medical image-to-image translation. It is based on the theory of "loss-correction". In RegGAN, the misaligned target images are considered as noisy labelsăand the generator is trained with an additional registration network to fit the misaligned noise distribution adaptively. The goal is to search for the common optimal solution to both image-to-image translation and registration tasks. We incorporated RegGAN into a few state-of-the-art image-to-image translation methods and demonstrated that RegGAN could be easily combined with these methods to improve their performances. Such as a simple CycleGAN in our mode surpasses latest NICEGAN even though using less network parameters. Based on our results, RegGAN outperformed both Pix2Pix on aligned data and Cycle-consistency on misaligned or unpaired data. RegGAN is insensitive to noises which makes it a better choice for a wide range of scenarios, especially for medical image-to-image translation tasks in which well pixel-wise aligned data are not available. Code and data used in this study can be found at https://github.com/Kid-Liet/Reg-GAN.

## 1 Introduction

Generative adversarial networks (GANs)[1] is a framework that simultaneously trains a generator $G$ and a discriminator $D$ through an adversarial process. The generator is used to translate the distribu-

---

[*]Equal contribution.

[†]Corresponding author.

35th Conference on Neural Information Processing Systems (NeurIPS 2021).

tion of source domain images $X$ to the distribution of target domain images $Y$. The discriminator is used to determine if the target domain images are likely from the generator or from the real data.

$$\min_G \max_D \mathcal{L}_{Adv}(G, D) = \mathbb{E}_y [log (D(y))] + \mathbb{E}_x [log (1 - D(G(x)))] \tag{1}$$

Supervised Pix2Pix[2] and unsupervised Cycle-consistency[3] are the two commonly used modes in GANs. Pix2Pix updates the generator ($G : X \rightarrow Y$) by minimizing pixel-level $L1$ loss between the source image $x$ and the target image $y$. Therefore, it requires well aligned paired images, where each pixel has a corresponding label.

$$\min_G \mathcal{L}_{L1}(G) = \mathbb{E}_{x,y} [\|y - G(x)\|_1] \tag{2}$$

Well aligned paired images, however, are not always available in real-world scenarios. To address the challenges caused by misaligned images, Cycle-consistency was developed which was based on the assumption that the generator $G$ from the source domain $X$ to the target domain $Y$ ($G : X \rightarrow Y$) was the reverse of the generator $F$ from $Y$ to $X$ ($F : Y \rightarrow X$). Compared to the Pix2Pix mode, the Cycle-consistency mode works better on misaligned or unpaired images.

$$\min_G \min_F \mathcal{L}_{Cyc}(G, F) = \mathbb{E}_x [\|F(G(x)) - x\|_1] + \mathbb{E}_y [\|G(F(y)) - y\|_1] \tag{3}$$

The Cycle-consistency mode, however, has its limitations. In the field of medical image-to-image translation, it requires not only the style translation between image domains, but also the translation between specific pair of images. The optimal solution should be unique. For example, the translated images should maintain the anatomical features of the original images as much as possible. It is known that the Cycle-consistency mode may produce multiple solutions[4, 5], meaning that the training process may be relatively perturbing and the results may not be accurate. The pix2pix mode is not ideal either. Even though it has a unique solution, it is difficult to satisfy the requirement asking for well aligned paired images. With misaligned images, the errors are propagated through the Pix2Pix mode which may result in unreasonable displacements on the final translated images.

As of today, there is no image-to-image translation mode that can outperform both the Pix2Pix mode on aligned data and the Cycle-consistency mode on misaligned or unpaired data. Inspired by[6–10], we consider the misaligned target images as noisy labels, which means that the existing problem is regarded as supervised learning with noisy labels. So we introduce a new image-to-image translation mode called RegGAN. Figure 1 provides a comparison of the three modes: Pix2Pix, Cycle-consistency and RegGAN. To facilitate reading, we summarize our contributions as follows.

- We demonstrate the feasibility of RegGAN from the theoretical perspective of "loss-correction". Specifically, we train the generator using an additional registration network to fit the misaligned noise distribution adaptively, with the goal to search for the common optimal solution for both image-to-image translation and registration tasks.
- RegGAN eliminates the requirement for well aligned paired images and searches unique solution in training process. Based on our results, RegGAN outperformed both Pix2Pix on aligned data and Cycle-consistencyăon misaligned or unpaired data.
- RegGAN can be integrated into other methods without changing the original network architecture. Compared to Cycle-consistency with two generators and discriminators, RegGAN can provide better performance using less network parameters.

## 2 Related Work

**Image-to-image Translation:** Generative adversarial networks (GANs) have shown great potential in the field of image-to-image translation[11–16]. It has been successfully implemented in medical image analysis like segmentation[17], registration[18, 19] and dose calculation[20]. The existing modes, however, have their limitations. Specifically, the Pix2Pix mode[2] requires well aligned paired images which may not always be available. The Cycle-consistency mode can achieve unsupervised image-to-image translation. With a Cycle-consistency loss, it can be used for misaligned images. Based on Cycle-consistency, many methods[3, 21–30] have been developed including CycleGAN[3] and its variants such as MUNIT[31] and UNIT[32] in which both image content and style information are used to decouple and reconstruct the image-to-image translation

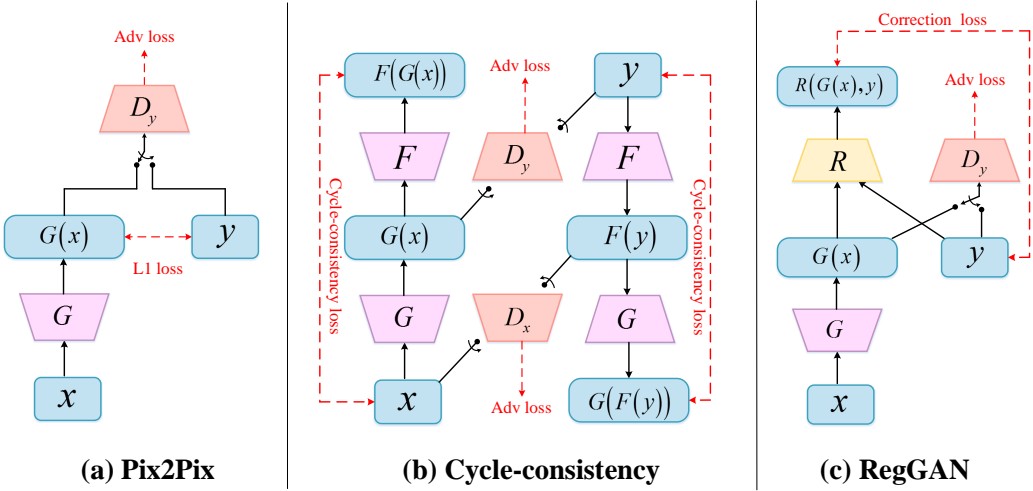

Figure 1: Comparison among the modes of Pix2Pix, CycleGAN and RegGAN.

task; U-gat-it[33] with a self-attention mechanism added; and NICEGAN[34] proposed to reuse the discriminator for encoding. The main limitation of Cycle-consistency is that it may produce multiple solutions and therefore is sensitive to perturbation, making it difficult to meet the high accuracy requirements of medical image-to-image translation tasks.

**Learning From Noisy Labels:** Neural network anti-noise training has made great progress. Current research are mainly focused on: estimating the noise transition matrix[7, 35–40], designing a robust loss function[41–44], correcting the noise label[45–50], sampling importance weighting[51–55] and meta-learning[56–59]. Our work is in the category of estimating the noise transition matrix. Compared to conventional noise transition estimation, we mitigate the issue and simplify the task by acquiring prior knowledge of noise distribution.

**Deformable Registration:** Traditional image registration methods have gained widespread acceptance, such as Demons[60], B-spline[61] and elastic deformation model[62]. One of the most popular deep learning methods is Voxelmorph[63]. In this study, a CNN model was trained to predict the deformable vector filed (DVF). The time-consuming gradient derivation process was thus skipped to improve the calculation efficiency. Affine registration and a vector momentum-parameterized stationary velocity field (vSVF)[64] was implemented to get better transformation regulation. Fast Symmetric method[65] used symmetric maximum similarity. Deep flash[66] outperformed other models in terms of training and calculation time.

Closest to our work, Arar.M et al[67] introduced a multi-modal registration method for natural images based on geometry preserving. But their work focused only on registration and did not demonstrate results of image-to-image translation or discuss the relationship between registration and image-to-image translation. The key insight of our work is that we demonstrated the feasibility of using registration to significantly improve the performance of image-to-image translation because the noise could be eliminated adaptively during the joint training process. What we propose in the paper is a completely new mode for medical image-to-image translation.

## 3 Methodology

### 3.1 Theoretical Motivation

If we considerămisaligned target images as noisy labels,ăthe training for image-to-image translation becomes aăsupervised learning processăwith noisy labels. Given a training dataset $\{(x_n, \widetilde{y}_n)\}_{n=1}^{N}$ with $N$ noisy labels in which $x_n$, $\widetilde{y}_n$ are images from two modalities and assume $y_n$ is the correct label for $x_n$, but it is unknown in real-world scenarios. Our goal is to train a generator using the dataset $\{(x_n, \widetilde{y}_n)\}_{n=1}^{N}$ with noisy labels and achieve the performance equivalent to trained on clean

dataset $\{(x_n, y_n)\}_{n=1}^{N}$ as much as possible. Direct optimization based on Equations 4 usually does not work and can lead to bad results because the generator cannot squeeze out the influence of noise.

$$\hat{G} = \arg\min_{G} \frac{1}{N} \sum_{n=1}^{N} \mathcal{L}\left(G\left(x_n\right), \widetilde{y}_n\right) \tag{4}$$

To address the noise issue, we propose a solution based on "loss-correction"[7] shown in Equations 5. Our solution corrects the output of the generator $G(x_n)$ by modeling a noise transition $\phi$ to match the noise distribution. Previously, Patrini et al[7] proved mathematically that the model trained with the noisy labels could be equivalent to the model trained with the clean labels, if the noise transition $\phi$ matches the noise distribution.

$$\hat{G} = \arg\min_{G} \frac{1}{N} \sum_{n=1}^{N} \mathcal{L}\left(\phi \circ G\left(x_n\right), \widetilde{y}_n\right) \tag{5}$$

To achieve this, Goldberger et al[36] proposed to view the correct label as a latent random variable and explicitly model the label noise as a part of the network architecture, denoted by $R$. Then, Equations 5 can be rewritten in the form of log-likelihood, which is used as the loss function for neural network training.

$$\begin{aligned} \mathcal{L}\left(G, R\right) &= -\sum_{n=1}^{N} log\left(p\left(\widetilde{y}_n | y_n; R\right) p\left(y_n | x_n; G\right)\right) \\ &= -\sum_{n=1}^{N} log\left(p\left(\widetilde{y}_n | x_n; G, R\right)\right) \end{aligned} \tag{6}$$

### 3.2 RegGAN

Compared to existing methods that use expectation-maximum[7, 36], fully connected layers[35], anchor point estimate[37] and Drichlet-distribution[38] to solve Equations 6. In our problem, the type of noise distribution is clearer, it can be expressed as displacement error: $\widetilde{y} = y \circ T$. Here $T$ is expressed as a random deformation field, which produces random displacement for each pixel. So we adopt a registration network $R$ after the generator $G$ as label noise model to correct the results. The Correction loss is shown Equations 7:

$$\min_{G, R} \mathcal{L}_{Corr}\left(G, R\right) = \mathbb{E}_{x, \widetilde{y}}\left[\|\widetilde{y} - G\left(x\right) \circ R\left(G\left(x\right), \widetilde{y}\right)\|_1\right] \tag{7}$$

where, $R\left(G\left(x\right), \widetilde{y}\right)$ is the deformation field and $\circ$ represents the resamples operation. The registration network is based on U-Net[68]. A smoothness loss[63] is defined in Equations 8 to evaluate the smoothness of the deformation field and minimize the gradient of the deformation field.

$$\min_{R} \mathcal{L}_{Smooth}\left(R\right) = \mathbb{E}_{x, \widetilde{y}}\left[\|\nabla R\left(G\left(x\right), \widetilde{y}\right)\|^2\right] \tag{8}$$

Finally, we add the Aversarial loss between the generator and the discriminator (Equations 1), and the total loss is expressed in Equations 9.

$$\min_{G, R} \max_{D} \mathcal{L}_{Total}\left(G, R, D\right) = \mathcal{L}_{Corr} + \mathcal{L}_{Smooth} + \mathcal{L}_{Adv} \tag{9}$$

## 4 Experiments

Performance evaluation of RegGAN was conducted through three investigations to 1) demonstrate the feasibility and superiority of the RegGAN mode in various methods, and 2) assess RegGANs sensitivity to noise, and 3) explore the availability of the RegGAN on unpaired data.

### 4.1 Dataset

The open-access dataset (BraTS 2018[69]) was used to evaluate the proposed RegGAN mode. The training dataset and testing dataset contained 8457 and 979 pairs of T1 and T2 MR images, respectively. BraTS 2018 was selected because the original images were paired and well aligned. We

created misaligned images by randomly adding different levels of rotation, translation and rescaling to the original images. And we randomly sample one image from T1 and the other one from T2 when training on unpaired images. The availability of well aligned paired images, misaligned paired images, and unpaired images allow us to evaluate the performances of all three modes (Pix2Pix, Cycle-consistency and RegGAN).

## 4.2   Performances in Different Methods

The primary motivation of introducing RegGAN was to address challenges caused by misaligned data. Therefore, in this section, misaligned data were used in model training to demonstrate the feasibility and superiority of RegGAN. We selected the most popular CycleGAN[3] and its variants MUNIT[31], UNIT[32], and NICEGAN[34] as the methods for evaluation and compared the following four modes for each method.

- **C(Cycle-consistency):** The most primitive mode of all methods, with Cycle-consistency loss (Equations 3) as the main constraint. Two generators and two discriminators are required in this mode.

- **C+R (Cycle-consistency + Registration):** The RegGAN mode is combined with the mode **C**. Registration network ($R$) and Correction loss (Equations 7) are added to the constraints.

- **NC(Non Cycle-consistency):** Only Adversarial loss (Equations 1) is used for updating. Compared to the mode **C**, Cycle-consistency loss is removed. Only one generator and one discriminator are required in this mode.

- **NC+R(Non Cycle-consistency + Registration):** A registration network ($R$) and Correction loss (Equations 7) are added to the mode **NC**. It is the proposed RegGAN mode.

Table 1: Comparison of CycleGAN, MUNIT, UNIT and NICEGAN using four training modes(C, C+R, NC and NC+R).

| Modes \ Methods Index | | CycleGAN | MUNIT | UNIT | NICEGAN |
|---|---|---|---|---|---|
| NMAE ↓ | C | 0.089 | 0.11 | 0.087 | 0.082 |
| | C+R | (-0.012)0.077 | (-0.022)0.088 | (-0.013)0.074 | (-0.011)**0.071** |
| | NC | 0.11 | 0.10 | 0.098 | 0.089 |
| | NC+R | (-0.038)**0.072** | (-0.021)**0.079** | (-0.027)**0.071** | (-0.019)**0.070** |
| PSNR ↑ | C | 23.5 | 20.6 | 24.6 | 25.2 |
| | C+R | (+0.3)23.8 | (+2.1)22.7 | (+0.7)25.3 | (+0.9)26.1 |
| | NC | 20.2 | 21.5 | 23.7 | 23.5 |
| | NC+R | (+5.4)**25.6** | (+2.3)**23.8** | (+1.8)**25.5** | (+2.8)**26.3** |
| SSIM ↑ | C | 0.83 | 0.80 | 0.84 | 0.83 |
| | C+R | (+0.02)0.85 | (+0.03) 0.83 | (+0.02)**0.86** | (+0.03)**0.86** |
| | NC | 0.79 | 0.81 | 0.83 | 0.84 |
| | NC+R | (+0.07)**0.86** | (+0.04)**0.85** | (+0.03) **0.86** | (+0.02)**0.86** |

To evaluate the performance of each method on misaligned data, we randomly added [-5, +5] degrees of angle rotation, [-5, +5] percent of translation, and [-5, +5] percent of rescaling to the original T1 and T2 images on the training dataset.

To ensure fair comparison, we used the same training strategy and hyperparameters for all methods and modes (see supplementary materials for details). The Normalized Mean Absolute Error (NMAE), Peak Signal to Noise Ratio (PSNR) and Structural Similarity (SSIM) were used as metrics to evaluate the performances of trained models based on the testing dataset. To avoid false high results of index, we excluded the image background from the calculation. Table 1 summarized the results for all methods and modes under the current investigation.

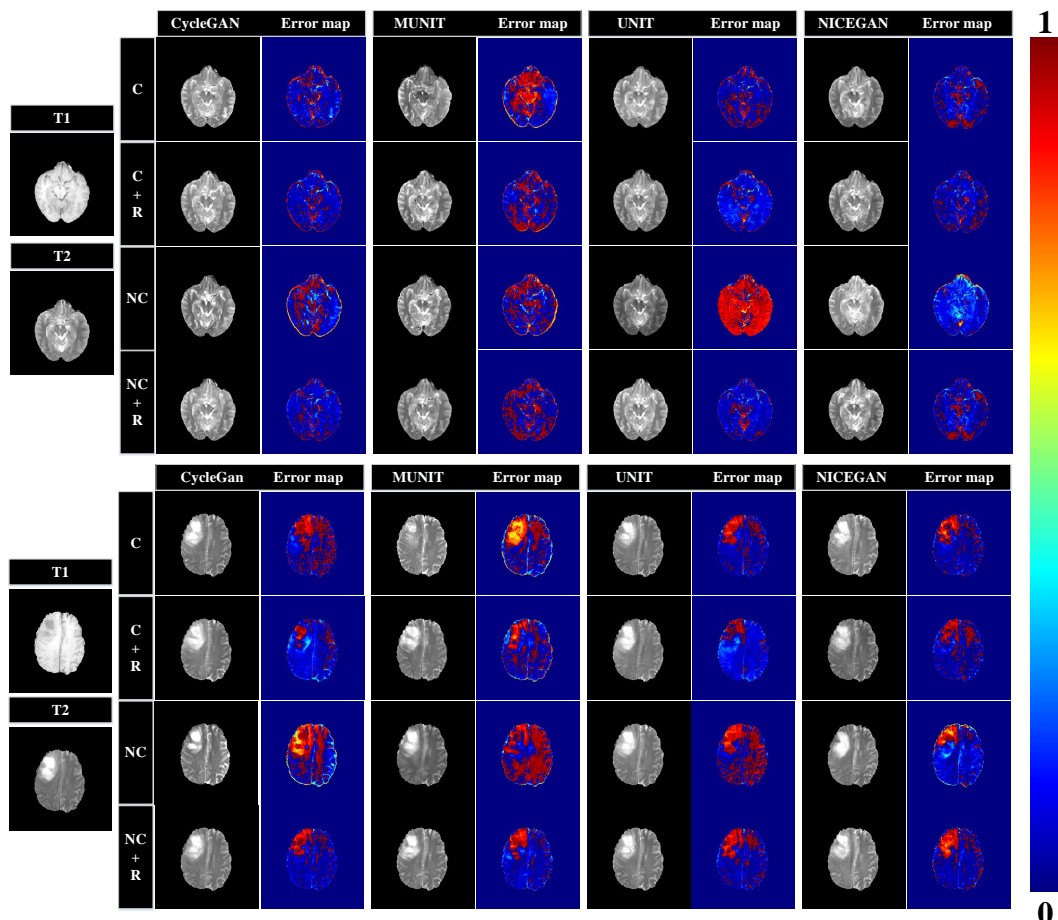

Figure 2: The errors of different modes in different methods.

Based on the results from the Table 1, we can reach several conclusions. First, adding the registration network (**+R**) significantly improves the performances of the methods. This is true for all methods in both **C** and **NC** modes. It clearly demonstrates that RegGAN can be incorporated in various methods or combined with different network architectures to improve the performances. Second, the **C** mode is in general better than the **NC** mode for most of methods. Adding the registration network (**+R**) improves the performance of the **NC** mode more than that of the **C** mode. In fact, our results show that the **NC+R** mode is even better than the **C+R** mode, implying that "Cycle-consistency loss" may play a negative role when it is combined with RegGAN. Compared with the commonly used **C** mode with two generators and two discriminators, RegGAN has fewer parameters but provides better performance. The simple CycleGAN method in the **NC+R** mode outperforms the current state-of-the-art method NICEGAN in the **C** mode by 0.01, 0.4, 0.03 for NMAE, PSNR and SSIM, respectively. The **NC+R** mode can also be used to improve the performance of NICEGAN. In fact, the performance of NICEGAN in the **NC+R** mode is the best among all combinations of the 4 methods and 4 modes.

Figure 2 shows representative results from various combinations of the 4 methods (CycleGAN, MUNIT, UNIT and NICEGAN) and 4 modes (**C**, **C+R**, **NC** and **NC+R**). For all aspects of the image (from the tumor areas and the details), the combinations that use the registration network (**+R**) always provide more realistic and accurate results than those that do not use the registration network (**+R**).

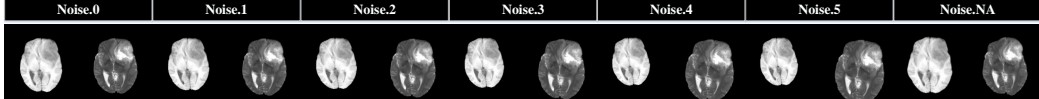

Figure 3: Example images at seven different levels of introduced noise.

Table 2: Comparison of the NMAE, PSNR and SSIM for CycleGAN(**C**), Pix2Pix and RegGAN under 7 levels of noise.

| | | Noise.0 | Noise.1 | Noise.2 | Noise.3 | Noise.4 | Noise.5 | Noise.NA |
|---|---|---|---|---|---|---|---|---|
| | Rotate | $0°$ | $±1°$ | $±2°$ | $±3°$ | $±4°$ | $±5°$ | ✗ |
| Setting | Translation | $0\%$ | $±2\%$ | $±4\%$ | $±6\%$ | $±8\%$ | $±10\%$ | ✗ |
| | Rescaling | $0\%$ | $±2\%$ | $±4\%$ | $±6\%$ | $±8\%$ | $±10\%$ | ✗ |
| | NMAE $\downarrow$ | 0.084 | 0.095 | 0.087 | 0.094 | 0.087 | 0.110 | 0.091 |
| CycleGAN(**C**) | PSNR $\uparrow$ | 23.9 | 22.5 | 23.7 | 23.3 | 23.9 | 23.7 | 23.5 |
| | SSIM $\uparrow$ | 0.83 | 0.83 | 0.82 | 0.81 | 0.82 | 0.79 | 0.83 |
| | NMAE $\downarrow$ | 0.075 | 0.103 | 0.139 | 0.161 | 0.175 | 0.181 | 0.086 |
| Pix2Pix | PSNR $\uparrow$ | 25.6 | 22.3 | 18.9 | 16.2 | 15.3 | 15.0 | 21.1 |
| | SSIM $\uparrow$ | 0.85 | 0.82 | 0.78 | 0.76 | 0.74 | 0.74 | 0.82 |
| | NMAE $\downarrow$ | **0.071** | **0.073** | **0.071** | **0.072** | **0.072** | **0.072** | **0.071** |
| RegGAN | PSNR $\uparrow$ | **26** | **25.6** | **25.9** | **25.7** | **25.4** | **25.2** | **25.9** |
| | SSIM $\uparrow$ | **0.86** | **0.86** | **0.86** | **0.86** | **0.86** | **0.85** | **0.86** |

## 4.3 Performances in Different Noise Levels

To evaluate the sensitivity of RegGAN to noise, we selected a simple network architecture.(CycleGAN) with the intention to minimize interference from other factors. The same network architecture was used for all three modes: CycleGAN(**C**), Pix2Pix and RegGAN. Seven levels of noise were used in the evaluation. Table 2 lists the specific noise setting and range for each noise level. Noise.0 means the original dataset with no added noise. Noise.5 is the highest level of noise. At Noise.5, the data are likely from different patients. In addition, we also made non-affine noise(Noise.NA) settings. The non-affine noise is generated by spatially transforming T1 and T2 using elastic transformations on control points followed by Gaussian smoothing. Figure 3 shows example images at different levels of introduced noise.

Table 2 lists the quantitative evaluation metrics from 3 modes at 7 levels of noise. It is clear that RegGAN outperforms CycleGAN(**C**) under all noise levels. Figure 4**(a)** shows the test results from each epoch during the training process for both RegGAN and CycleGAN(**C**). Curves of different colors corresponds to different levels of noise. We notice that CycleGAN(**C**) is not very stable during the training process. The test results fluctuate significantly and cannot converge well. This may be caused by the fact that the solution of CycleGAN (**C**) is not unique. As a comparison, RegGAN is quite stable. Although the results from different levels of noise may vary at the beginning of training, all curves converge to a similar result after multiple epoches of training, indicating that RegGAN is more robust to noise compared to CycleGAN (**C**).

Based on Table 2, we notice that the performance of Pix2Pix deteriorates rapidly as the noise increases. This is as expected because Pix2Pix requires well aligned paired images. Surprisingly, the performances of RegGAN at all noise levels exceed those of Pix2Pix with no noise. Figure 4**(b)** shows the test results at each epoch of RegGAN and Pix2Pix under Noise.0 (i.e., no noise). Theoretically, the performances of RegGAN and Pix2Pix should be similar on perfectly aligned paired datasets because the registration network of RegGAN does not help and RegGAN is equivalent to

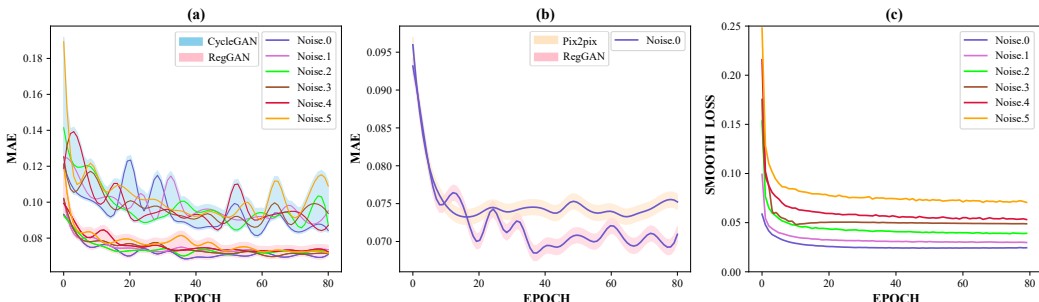

Figure 4: Quantitative evaluation metrics at different epochs in the training process. **(a)** Comparison of CycleGAN and RegGAN at different levels of noise. **(b)** Comparison of Pix2Pix and RegGAN at Noise.0 (i.e., no noise). **(c)** RegGAN's Smoothness loss under different levels of noise.

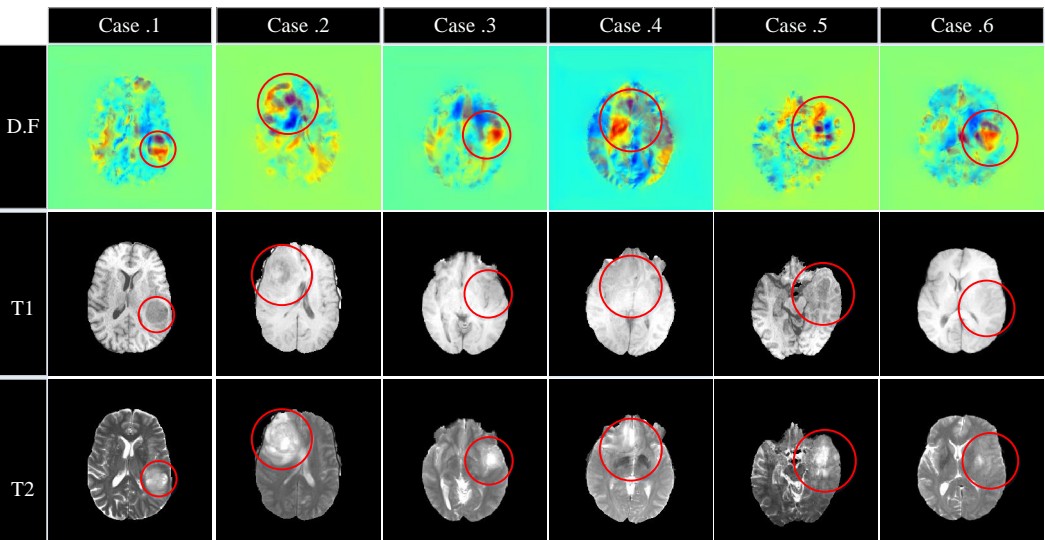

Figure 5: The misalignment of orginal image pairs and corresponding deformation fields.

Pix2Pix. A possible explanation to our results is that in the medical field, the perfectly pixel-wise aligned dataset may not practically exist. Even for BraTS 2018[63] which is recognized as well aligned, it is still possible that there exists slight misalignment. As a result, adding the registration network is always likely to improve the performances in real-world scenarios. To verify our explanation, we plotted the Smoothness loss of RegGAN under different noise levels as shown in Figure 4(c). Large Smoothness loss corresponds to large deformation field displacement. First, we notice that the Smoothness loss under Noise.0 never completely goes to 0, indicating the existence of misalignment and potential usefulness of the registration network. Second, the noise level and Smoothness loss show a step-like positive correlation, which means that RegGAN can adaptively handle the noise distribution, i.e., the registration network can determine the range of deformation according to the noise level. In addition, we can see that even under the setting of non-affine noise, the above conclusion still holds. Because what the registration network corrects is deformation noise.

we also show some original image pairs and visualize the corresponding deformation fields output by registration network in Figure 5. Obviously, there is some misalignment between the original T1 and T2 images, and such misalignment is represented by the deformation fields (highlighted by red circle).

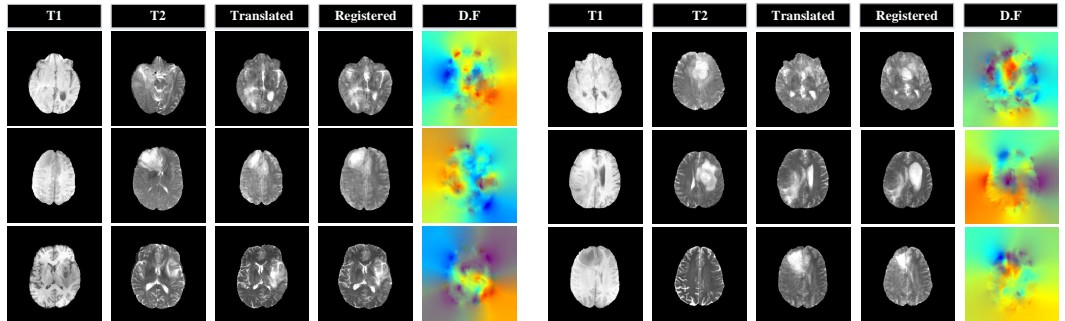

| | Index | NMAE↓ | PSNR↑ | SSIM↑ |
|---|---|---|---|---|
| Mode | | | | |
| Pix2Pix | | 0.180 | 15.5 | 0.71 |
| CycleGAN(**C**) | | 0.094 | 23.6 | **0.83** |
| RegGAN | | **0.086** | **24.0** | 0.83 |

Figure 6: Performance comparison of the three modes (CycleGAN(**C**), Pix2Pix and RegGAN) on unpaired dataset.

Figure 7: Display of RegGAN's output on unpaired data. **T1** and **T2** are unpaired images. The **Translated** represents the translation result of T1 to T2. **Registered** represents the registration result of the translated images. **D.F** represents deformation fields.

## 4.4 Performances on Unpaired Dataset

So far, our investigations are based on paired datasets. We also want to explore how RegGAN performs using unpaired datasets. In practice, this is not recommended because even different patients may have similarities in their body tissues of adjacent layers. For unpaired datasets, we can conduct rigid registration first in 3D space and then use RegGAN for training. Unpaired data can be treated as having larger scale noise. If the correction capability is strong enough, RegGAN can still work effectively. The comparison of the performances of three modes on the unpaired dataset is shown in Figure 6.

With unpaired datasets, Pix2Pix no longer considers the characteristics of the input T1 images and thus has the worst performance. Due to the challenges in fitting the noise, the performance improvement from replacing CycleGAN(**C**) with RegGAN using unpaired datasets may not be as dramatic as that demonstrated using paired datasets, but RegGAN still has the best performance under unpaired conditions. In Figure 7, we show some examples of how RegGAN corrects noise on unpaired dataset. It can be seen that RegGAN will try its best to eliminate the influnce of noise through registration.

Based on our results, it is reasonable to reach the conclusions below. In all circumstances, RegGAN demonstrates better performance compared to Pix2Pix and CycleGAN(**C**).

- For paired and aligned conditions, RegGAN ≥ Pix2Pix > CycleGAN(**C**).

- For paired but misaligned conditions, RegGAN > CycleGAN(**C**) >Pix2Pix.

- For unpaired conditions, RegGAN > CycleGAN(**C**) >Pix2Pix.

## Conclusion

In this study, we introduced a new image-to-image translation mode RegGAN to the medical community that can break the dilemma of image-to-image translation task. Using a public BraTS 2018 dataset, we demonstrated the feasibility of RegGAN and its superior performance compared to Pix2Pix and Cycle-consistency. We validated that RegGAN can be incorporated into various existing methods to improve their performances. We also evaluated the sensitivity of RegGAN to noise. Our results confirmed that RegGAN could adapt well to various scenarios from no noise to large-scale noise. The superior performance of RegGAN makes it a better choice over Pix2Pix and Cycle-consistency whether datasets are aligned or not. However, this mode may not work well on natural images. The noise may cannot be considered simply as deformation errors due to the differences in natural images are much greater than those in medical images.

## Broader Impact

Image-to-image translation has been one of the main focuses in medical image analysis, as it aids in diagnosis and treatment. Previously, physicians had to use different medical imaging equipments if they wanted to get different image sequences of a patient, which was time-consuming and expensive. Pix2Pix mode is expected to solve this problem by its outstanding performance in image-to-image translation. In most of clinical scenarios, however, it is not practical to create such a large well aligned dataset for Pix2Pix mode. Cycle-consistence mode does not need well aligned dataset but can not meet the high-precision requirements of medical image analysis. Our work aims to provide a general image-to-image translation mode, which not only has no strict requirements on the dataset, but also can meet the clinical requirements in terms of image quality. In the future, we will attempt to obtain multi-modal dataset(eg MR-CT) for clinical verification. We foresee positive impacts if the mode is applied to diagnosis in radiology, treatment planning and research.

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
