# Breaking the Dilemma of Medical Image-to-image Translation

**Lingke Kong**[*]
Manteia Tech
konglingke@manteiatech.com

**Chenyu Lian**[*]
Xiamen University
cylian@stu.xmu.edu.cn

**Detian Huang**
Huaqiao University
huangdetian@hqu.edu.cn

**Zhenjiang Li**
Shandong University
zhenjli1987@163.com

**Yanle Hu**[†]
Mayo Clinic Arizona
Hu.Yanle@mayo.edu

**Qichao Zhou**[†]
Manteia Tech
zhouqc@manteiatech.com

## A    Training Details

All the experiments were implemented in Pytorch software on 64-bit Ubuntu Linux system with 96GB RAM and 24GB Nvidia Titan RTX GPU. All the images were normalized to [-1, 1] and resampled to 256×256. We train all methods using the Adam optimizer with the learning rate of 1e-4 and $(\beta_1, \beta_2)$ = (0.5, 0.999). The batch size was set to 1 with weight decay 1e-4. The training process includes totally 80 epochs and over 640K iterations. We also set different weights for different loss functions, as shown in Table 1.

Table 1: The different weights for different loss functions.

| Loss / Weight | $\mathcal{L}_{adv}$ | $\mathcal{L}_{L1}$ | $\mathcal{L}_{cyc}$ | $\mathcal{L}_{corr}$ | $\mathcal{L}_{smooth}$ |
|---|---|---|---|---|---|
| $\lambda$ | 1 | 100 | 10 | 20 | 10 |

**CycleGAN** uses two downsampling convolution blocks, nine residual blocks, two up-sampling deconvolution blocks and four discriminator layers. Codes are on `https://github.com/junyanz/pytorchCycleGAN-and-pix2pix`.

**MUNIT** assumes that the image representation can be decomposed into a content code and a style code. It splits a generator into two encoders and a decoder. Codes are on `https://github.com/NVlabs/MUNIT`.

**UNIT** assumes that the different modality of an image should share the same latent coding space. It shares the weight of the high-level layer stage of the encoder and decoder. Codes are on `https://github.com/mingyuliutw/UNIT`.

---

[*]Equal contribution.
[†]Corresponding author.

35th Conference on Neural Information Processing Systems (NeurIPS 2021).

**NICEGAN** reuses discriminators for encoding specifically for unsupervised image-to-image translation. By such a reusing, a more compact and more effective architecture is derived. Codes are on https://github.com/alpc91/NICE-GAN-pytorch.

# B  Theoretical Analysis

In this section, we will analyze the generalization form for the proposed mode.

**Definition:** In the image-to-image translation task, the goal is to optimize and get a generator $G$ : $\arg\min_{G} \mathbb{E}_{x\,y}\mathcal{L}(G(x), y)$. Where $(x, y)$ are paired and aligned multi-modal images, $\forall x, y \in \mathbb{R}^{H \times W}$ image space. $\mathcal{L}$ is the loss function. But in practice, we can only get noisy labels $(x, \widetilde{y})$, and the correct label $y$ is unknown. The relationship between $\widetilde{y}$ and $y$ can be expressed as displacement error: $\widetilde{y} = y \circ T$. Here $T$ is expressed as a random deformation field, which produces displacement for each pixel, $\forall T \in \mathbb{R}^{2 \times H \times W}$. If we can build an unbiased estimator of model $R$, it can fit the noise distribution $T$ well, such that under expected label noise the corrected loss equals the original one computed on clean data.

**Theorem 1.** *Suppose the deformation field $T$ is smooth enough for: $T \circ T^{-1} \equiv I$. $I$ represents identical transformation. Then, the minimizer of the corrected loss under the noisy distribution is the same as the minimizer of the original loss under the clean distribution:*

$$\arg\min_{G} \mathbb{E}_{x\,\widetilde{y}}\mathcal{L}(G(x) \circ T, \widetilde{y}) = \arg\min_{G} \mathbb{E}_{x\,y}\mathcal{L}(G(x), y) \tag{1}$$

*Proof.* Put $\widetilde{y} = y \circ T$ in the left-hand side of the above equation:

$$
\begin{aligned}
\arg\min_{G} \mathbb{E}_{x\,\widetilde{y}}\mathcal{L}(G(x) \circ T, \widetilde{y}) &= \arg\min_{G} \mathbb{E}_{x\,y}\mathcal{L}(G(x) \circ T, y \circ T) \\
&= \arg\min_{G} \mathbb{E}_{x\,y}\mathcal{L}(G(x \circ T^{-1}) \circ T, y \circ T \circ T^{-1}) \\
&= \arg\min_{G} \mathbb{E}_{x\,y}\mathcal{L}(G(x) \circ T \circ T^{-1}, y \circ T \circ T^{-1}) \\
&= \arg\min_{G} \mathbb{E}_{x\,y}\mathcal{L}(G(x), y)
\end{aligned}
\tag{2}
$$

$\square$

# C  More Rusults

The results in Section 4.3 show that RegGAN mode is superior to Pix2Pix mode at all noise levels. Here, we give a comparison of error mappings of Pix2Pix and RegGAN under Noise.0(Figure 1) and Noise.5(Figure 2) respectively. In Figure 8, the RegGANs result has smaller errors and smoother texture details compared to Pix2Pix, but the overall difference is not significant. In Figure 9, Pix2Pix is no longer effective, while RegGAN maintains a good performance.

Figure 1: Qualitative results under Noise.0 (o noise). We show eight examples, which are randomly selected from total test dataset.

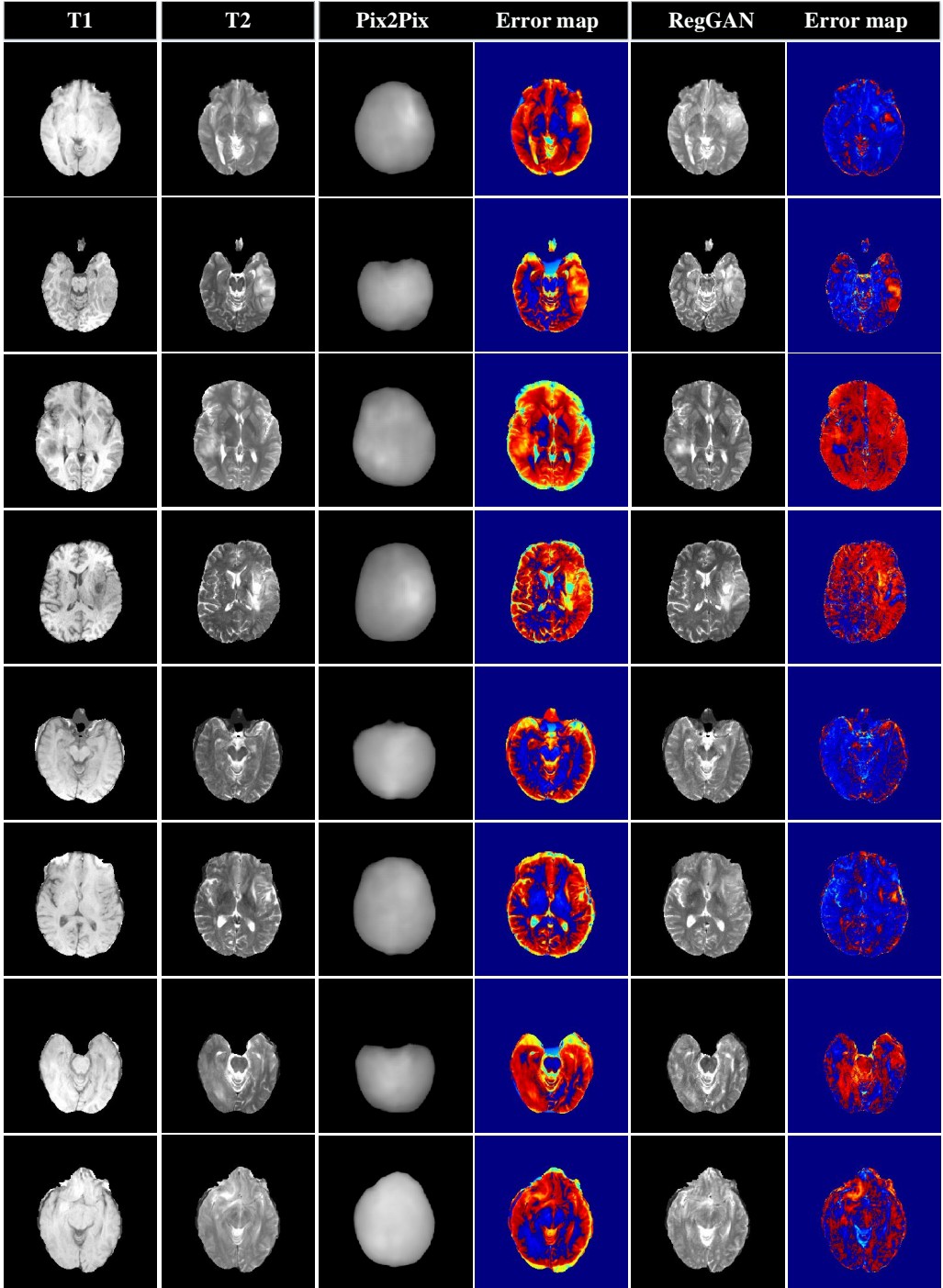

Figure 2: Qualitative results under Noise.5(largest noise in our experiments). We show eight examples, each one is randomly selected from total test dataset.