# OpenReview forum: "Breaking the Dilemma of Medical Image-to-image Translation"
_NeurIPS.cc/2021/Conference — NeurIPS 2021 Spotlight_

### Official Review · Reviewer_yp9h · 2021-07-14

**Rating:** 6
**Confidence:** 3

**Summary:**

This paper considers the setting where paired images are available but some of them are misaligned. To address this issue, it proposes to add a registration network after the translation network as the "correction loss".

**Limitations And Societal Impact:**

Yes

**Main Review:**

Pros

1. The setting is interesting. Pix2pix needs well-paired data while CycleGAN fails to utilize some paired data. Instead, this paper considers the noisy case where some paired data are misaligned.

2. Extensive experiments are conducted and show the performance of the propose method.


Cons:

1. The novelty is limited.  This paper introduces a registration network after the translation network to address the misaligned data. But a published noisy label paper [7] already introduces a noise transition $\phi$. This paper adapts it to medical image translation and uses a registration network.  The theoretical motivation is also borrowed from [7].

After reading authors' rebuttal, I think the technical novelty of this method is enough though more analysis (theoretical and more complex datasets) are needed. So I raised my score to 6.

Suggestions:

1. Authors may consider adding a brief introduction to registration network and deformation field before section 3.2, which may cause some confusion for people who are not in medical fields.

2. It would be better if authors can provide some theoretical analysis about this method. It is still not very clear why introducing a registration network can help align the data without any other specific regularization.


**Time Spent Reviewing:**

2

---

> ### Author Response · Authors · 2021-08-09
> **Reply to comments**
>
> Thank you very much for your comments and suggestions. We would like to use this opportunity to explain the innovation aspect of our work. Hopefully, it may help clear up the confusion.
>
> Q1: The novelty is limited.
>
> A2: The most innovative aspect of our work is that it proposed a completely new mode for medical image-to-image translation, and demonstrated the feasibility of using registration to significantly improve the performance of image-to-image translation because the noise could be eliminated adaptively during the joint training process. Based on our results, RegGAN outperformed both Pix2Pix on aligned data and Cycle-consistency on misaligned or unpaired data. Our work solves the problem of data misalignment in image translation tasks. The noise transition ϕ in our work represents the deformation field. Literature [7], on the other hand, addresses incorrect labels in image classification tasks. In their work, the noise transition ϕ represents the probability of label classification errors. Even though both involve the theory of noise transition, they belong to different fields, have different meanings, and intend to solve different problems.
> In fact, after the publication of the literature [7], a lot of work has been published and considered novel even in the area of classification tasks. Below are a few examples. Considering that our work provides an effective solution to a previously unresolved problem, we believe it is novel and could have significant impact to many fields such as clinical practice of radiation oncology.
>
> (1)Arazo E , Ortego D , Albert P , et al. Unsupervised Label Noise Modeling and Loss Correction[J]. PMLR-2019.
>
> (2)Han J , Luo P , Wang X . Deep Self-Learning From Noisy Labels[J]. EEE/CVF International Conference on Computer Vision(ICCV).2019
>
> (3)J Li, Wong Y , Zhao Q , et al. Learning to Learn From Noisy Labeled Data[C]// 2019 IEEE/CVF Conference on Computer Vision and Pattern Recognition (CVPR).
>
> (4)D. Hendrycks, M. Mazeika, D. Wilson, and K. Gimpel, “Using trusted data to train deep networks on labels corrupted by severe noise,” in Proc. NeurIPS, 2018, pp. 10 456–10 465
>
> (5)Yao Y, Liu T, Han B, et al. Dual t: Reducing estimation error for transition matrix in label-noise learning[C/OL]//Larochelle H, Ranzato M, Hadsell R, et al. Advances in Neural Information 279  Processing Systems: volume 33. Curran Associates, Inc., 2020: 7260-7271.
>
> (6)Are anchor points really indispensable in label noise learning?[C/OL]//NeurIPS.2019: 6835-6846.
>
> Suggestion1: Based on your suggestion, we added a brief introduction regarding registration. In addition, we mentioned the registration network as the architecture of U-Net in line 116 and cited the literature [62] for the source of the registration method. It is worth mentioning that registration, as a traditional basic task, has applications not only in the medical field but also in many other fields such as remote sensing, point cloud, navigation, tracking and detection.
>
> Suggestion2: Regarding your suggestion related to theoretical analysis, we have more details in the supplemental material. If you meant regularization as a constraint on the deformation field, it is in the formula (8) as a regularization factor for the smoothness of the deformation field.

---

> > ### Comment · Reviewer_yp9h · 2021-08-17
> > **Response**
> >
> > Thanks for your reply.
> >
> > I agree that this setting can be useful to medical image translation. So I raised my score to 5.
> >
> > But the technical novelty of this method is still limited to me. I read the published papers  provided in the comments but they are different from [7]. For example, Arazo et al proposed to model the noise and clean data as a mixture and learn the model based on the observation that random labels takes longer to learn than the clean labels.  In [7]., the forward correction loss is defined as $l(y, \phi(h(x)))$. The RegGAN loss is defined as $l(y, G(x)\circ R(G, y))$. These two loss functions are close though in different fields.

---

> > > ### Author Response · Authors · 2021-08-20
> > > **Reply to comments 2**
> > >
> > > We greatly appreciate that you recognized the value of our work and was willing to improve the score accordingly.
> > >
> > > Regarding the technical novelty, we would like to further clarify the difference between our work (RegGAN) and those by Patrini et al and Arazo et al.
> > >
> > > Patrini et al established the basic theory, stating that given ideal noise correction, training using the noisy data would achieve equivalent results compared to training using the clean data. In their theory, the key component is the noise correction method, or specifically solving the probability of label classification errors $\phi$.
> > >
> > > For tasks involving noise correction, the term corresponding to the noise correction loss can be generally denoted as $\mathcal{L}\left(y,\phi\left(h\left(x\right)\right) \right)$. It is the exact form of $\phi$ that differentiates various methods and grants novelty to each individual method. For example, Arazo et al improved the noise correction by “modeling the noise and clean data as a mixture and learning the model based on the observation that random labels took longer to learn than clean labels”, as mentioned in your comments. Arazo et al stated in their paper that “Finally, we obtain the probability of a sample being clean or noisy through the posterior probability”. Then they did loss correction using equation (10) and (13) in their paper, those were a specific form of $\mathcal{L}\left(y,\phi\left(h\left(x\right)\right) \right)$. In our work, we proposed a noise correction method for image-to-image translation utilizing the associated image registration. Our method uses L1-norm in the calculation, instead of cross entropy as in Patrini et al and Arazo et al’s work. The $\phi\left(h\left(x\right)\right)$ in our work represents the resampling $h\left(x\right)$ by $\phi$, whereas the $\phi\left(h\left(x\right)\right)$ in Patrini et al’s work is multiplication of $\phi$ and $h\left(x\right)$. Therefore, our proposed noise correction method is indeed different from those proposed by Patrini et al and Arazo et al, even though Equation (7) in our work also is the special form of $\mathcal{L}\left(y,\phi\left(h\left(x\right)\right) \right)$.
> > >
> > > The relationship between the three methods are summarized in the below table.
> > > $$
> > > \begin{array}{c|c|cc}
> > > \hline
> > > & Task&\text{How to get an ideal noise correction method}\\\\
> > > \hline
> > > & Classification & \text{Arazo(fit a beta mixture model to the loss of each sample and label noise)}\\\\
> > > \text{Patrini(basic theory)}&  & &   \\\\
> > > & Translation &\text{RegGAN(traind with an additional registration network to fit the noise)} \\\\
> > > \hline
> > > \end{array}
> > > $$
> > > $$
> > > \begin{array}{c|c|cc}
> > > \hline
> > > \text{Patrini(General Form)}& Method&\text{Special Form}\\\\
> > > \hline
> > > & Arazo & \mathcal{L}=-\sum_{i=1}^N \left(\left(1-w_{i}\right)y_{i} +w_{i}z_{i} \right)^{T}log\left(h_{i}\right)\\\\
> > > \mathcal{L}\left(y,\phi\left(h\left(x\right)\right) \right)&  & &   \\\\
> > > & RegGAN &\mathcal{L}=\sum_{i=1}^N\left(\|y_{i}-G\left(x_{i}\right)\circ R\left(G\left(x_{i}\right),y_{i}\right)\|_{1}\right) \\\\
> > > \hline
> > > \end{array}
> > > $$

---

> > > > ### Comment · Reviewer_yp9h · 2021-09-03
> > > > **about the novelty**
> > > >
> > > > Thanks for your clarification. It partly addressed my concern. So I raised my score to 6.

---

### Official Review · Reviewer_sQmz · 2021-07-15

**Rating:** 7
**Confidence:** 4

**Summary:**

The authors present a new medical image-to-image generative model, called RegGAN, that combines an adversarially trained image generator with a registration module that enforces alignment between the generated image and ‘noisy’, (un)paired ground truth image. The aim of the network is to search for the single optimal solution to both image-to-image translation and registration tasks. The authors compared their proposed method against two other general model types, Pix2Pix (paired aligned data) and Cycle GAN (unpaired or misaligned). For training and testing, they use the BRATS dataset, which consists of brain tumor MRIs, and translate T1 to T2 images. The authors were able to show better performance on all major metrics for both paired and unpaired data. They showed that RegGAN was more resistant to label noise and increased stability during training.

**Limitations And Societal Impact:**

The authors have restricted the noise to affine deformations only. RegGAN is capable of doing non-affine noise correction and these deformations may be encountered in clinical medicine (e.g. 1.5 vs 3 Tesla MRIs). The authors are encouraged to perform these experiments and add them to Table 2.

The current study is limited to a single image-to-image translation task: T1 to T2. While I understand that the authors are limited by the availability of open-source datasets, this task is highly artificial and generally would not occur in clinical medicine. It seems a better test, especially for demonstrating superior performance on unpaired or misaligned datasets, would be cross-modality, such as CT to MRI or vice versa. The authors state ‘Previously, physicians had to use different medical imaging equipment if they wanted to get multi-modal images of a patient, which was time-consuming and expensive... Our work aims to..meet the clinical requirements in terms of image quality’. No multi-modal testing is shown in the paper. I would either attempt to obtain a CT-MRI dataset (this would strengthen the impact of the results), or remove the previous statement.

Line 235: BraST > BraTS

Overall, good paper and a thoughtful contribution.


**Main Review:**

Originality: an original combination of a registration loss to a GAN objective in a medical image setting.
Quality and Clarity: good quality, the paper is well written and the figures are excellent.
Significance: Good contribution to a medical imaging problem that needs to be addressed.

**Time Spent Reviewing:**

2

---

> ### Author Response · Authors · 2021-08-09
> **Reply to comments**
>
> Thank you for your positive comments and suggestions.
>
> Q1: Add the comparison of non-affine noise.
>
> A1: The reviewer is absolute right that in clinical medicine, non-affine deformation is often encountered. And according to your suggestion, we made non-affine noise settings and added the results of experiments in Table 2. The non-affine noise is generated by spatially transforming one of the modality (T1) using elastic transformations on control points followed by Gaussian smoothing. Given RegGAN is capable of handling non-affine noise correction, it is expected that RegGAN can be more advantageous compared to Pix2Pix and CycleGAN.
>
> Q2: Add cross-modality testing.
>
> A2: We fully agree with your suggestion that it is better to perform experiments using realistic datasets such as MRI to CT translation. But as the reviewer pointed out, we were limited by the availability of open-source datasets. From the perspective of demonstrating the feasibility of RegGAN, the BraTS dataset was sufficient as shown in the zero-noise case in Table 2. To demonstrate the full advantage of RegGAN, we already started the collaboration with multiple hospitals. Currently, we are in the process of acquiring clinical datasets focusing on several disease sites such as head, abdomen and pelvis. As we know, clinical data acquisition can be very time consuming. But we are doing exactly what the reviewer suggested. We hope that we will be able to demonstrate the superior performance of RegGAN in real clinical scenarios soon.
>
> Q3: The typo and inaccuracy in the paper.
>
> A3: Also, we would like to thank the reviewer for pointing out the typo and inaccuracy of the statement in the paper. We have revised those according to your suggestions.

---

> > ### Author Response · Authors · 2021-08-13
> > **Experimental results of adding non-affine noise**
> >
> > According to your suggestion, the results of adding non-affine noise have been shown in the table below, and will be added to the revised paper later.
> >
> > CycleGAN(C)     MAE:0.091   PSNR:23.5    SSIM:0.83
> >
> > Pix2Pix               MAE:0.086   PSNR:21.1    SSIM:0.82
> >
> > RegGAN            MAE:0.071   PSNR:25.9      SSIM:0.86

---

### Official Review · Reviewer_REoq · 2021-07-15

**Rating:** 7
**Confidence:** 4

**Summary:**

The paper introduces a new method for image-to-image translation, which accommodates misalignment between training image pairs - a common problem particularly with medical images where motion or distortion often leads to misalignment that cannot be perfectly corrected through a-priori image registration.  The authors rightly state that this is a key missing component of existing image-to-image translation approaches such as pix2pix-type algorithms that assume matched pairs with perfect alignment, whereas cycle-consistency approaches do not exploit matched pairs when available.



**Ethical Concerns:**

Marked as N/A - the data is publicly available, so I guess that's fine.

**Limitations And Societal Impact:**

yes

**Main Review:**

I am enthusiastic from the start, as this is a problem I have encountered and thought about, and have been waiting for a nice solution!  I'm amazed no-one has done this before actually, but I am not aware of prior work on this issue.

The implementation the authors propose is simple but sensible and the right place to start with solving this important problem.  They simply add estimation of a deformation field that is learned concurrently with the image-to-image translation during training, as well as a smoothing loss to keep the deformation under control.

Experiments use the BRATS brain-MRI data set and add artificial misalignments to pairs of images with different MR contrast before learning image-to-image translations between those contrasts from the paired data set.  As expected, the proposed technique outperforms existing techniques that do not consider misalignment. The authors go on to show that the proposed technique also has benefit in situations where matched pairs are not available but simply unmatched examples of both contrasts even though the method is not designed for such situations.

My key criticism is that the experiments are rather artificial as the imposed misalignments are much larger than one would normally find between different MR contrasts.  Rotations scalings and displacements of the type the authors use to corrupt the alignment would be easily fixed by a-priori image registration. In practice the real problem arises from small residual misalignments after a-priori image registration - that tends to lead to blurring in estimated images as a consequence of learning from misaligned data sets. A fairer comparison with existing methods and more convincing demonstration of the benefits of the proposed technique would have come from registering the paired images prior to training, as that is what users of say pix2pix would naturally normally do.  We do get a hint of this in section 4.3 where the evaluation as a function of "noise level" (which here refers to the degree of alignment corruption) includes zero noise, ie no artificial misalignment but only residual misalignment after the registration step used in the original BRATS data.  The proposed technique does still show benefit in this "zero noise" test, which is encouraging, but I would have liked to see a more complete evaluation and comparison against the state of the art in this more realistic scenario.

Nevertheless, I remain enthusiastic about the proposed method and look forward to seeing more complete demonstrations of its capability as well as further refinement of the idea.

**Time Spent Reviewing:**

2

---

> ### Author Response · Authors · 2021-08-09
> **Reply to comments**
>
> We are honored to get your approval and we are greatly encouraged by your comments!
>
> Q1: The experiments are artificial.
>
> A1: We fully agree with your point of view. it is the residual misalignment after a-priori image registration that determines the real-life performance of each method. This was explored in Section 4.3. The case of zero noise in Table 2 corresponded to the scenario of registering the paired images prior to training. Our results did demonstrate the advantage of RegGAN. We greatly appreciate that the reviewer noticed this and recognized the significance of it. Compared to Pix2Pix with a-priori image registration, the advantage of RegGAN is likely to be greater than what was demonstrated because the BraTS dataset used in this study is an ideal dataset and has small residual misalignment. The BraTS dataset was selected because it was publically accessible. But the small residual misalignment in BraTS does provide a bias favorable towards Pix2Pix. In realistic scenarios (e.g., translation between MRI and CT, abdominal datasets, etc), the residual misalignment can be much larger. As Table 2 shows, the performance of Pix2Pix deteriorates rapidly with increased noise. The performance of RegGAN, on the other hand, is very robust to noise. Therefore, a greater advantage is expected by using RegGAN, instead of Pix2Pix, in realistic scenarios.The main focus of this work is to demonstrate the feasibility of RegGAN which uses registration to significantly improve the performance of image-to-image translation. Validating advantages of RegGAN within a wide scope of realistic scenarios is on our roadmap. In fact, we are collaborating with multiple hospitals to acquire datasets focused on disease sites like head, abdomen and pelvis. We hope that our work can inspire others to come up with more sophisticated medical image translation methods.

---

### Official Review · Reviewer_TPfs · 2021-07-19

**Rating:** 6
**Confidence:** 4

**Summary:**

The authors proposed a novel method for conducting image-to-image translation integrating registration network (RegGAN), and demonstrated the superior performance in T1 to T2 MRI image translation to the current methods based on Pix2Pix and cycle-consistency techniques.
The RegGAN performed slightly better than Pix2Pix method even without added noise likely due to inherent misalignment from the image acquisition. While the Pix2Pix method deteriorates rapidly with the added noise, RegGAN was robust against noise similarly to CycleGAN (with better performance).
The authors also experimented to train them with unpaired images, and RegGAN could generate T2 images more similar to the actual images than the other two methods.

**Limitations And Societal Impact:**

The paper still have a few limitations.
First, more traditional image (co)registration technique combined with Pix2Pix technique hasn't been explored. Especially in paired setting with small or zero noise, Pix2Pix followed by rigid alignment may perform similarly to RegGAN. While registering T2 images to T1 images can be challenging in some cases, another approach would be register both T1 and T2 images to reference T1 and T2 images, respectively at first and train Pix2Pix translation between the registered images. I still think the end-to-end approach of the authors have benefits over these in terms of simplicity, it would be nice to see such comparisons.
Second, the performance metric used is mostly based on image-level, and doesn't reflect actual utility in medical diagnosis. Slight improvement in aligning the normal tissues can improve these metrics, while not improving the clinical utility. Potentially, even with the higher score, the translated images may completely miss important abnormalities which is normally highlighted in T2 images (e.g. high: ischemic stroke, low: hemorrhage). This can be in particular problematic in unpaired settings. At least some examples should be included.

**Main Review:**

Originality:
While using deep learning network for medical image registration is not new (e.g. BIRNet), combining it with generator network for image-to-image translation and training it end-to-end is the original approach.

Quality and clarity:
The paper is clearly written with enough details.

Significance:
T1 and T2 MRI images are typically well aligned, and the performance gain in paired setting is not large. Still, the results were presented well to indicate the potential to be applied to other medical imaging tasks where distortion from motion artifacts are more evident.

**Time Spent Reviewing:**

2 hrs

---

> ### Author Response · Authors · 2021-08-09
> **Reply to comments**
>
> Thank you for your supportive and insightful comments.
>
> Q1: The traditional image (co)registration technique combined with Pix2Pix technique hasn't been explored.
>
> A1: The innovative aspect of our work is to show that the joint registration method in the process of image translation can effectively correct noise and bring meaningful performance improvement. Even though we didn’t specifically state it, combining traditional image registration with Pix2Pix was indeed explored in the current work. In Table 2, the noise level of 0 corresponds to the suggested case of training Pix2Pix translation between the registered images because the BraTS2018 dataset is well aligned. Our results show that RegGAN outperformed Pix2Pix even for datasets with registered images. The results are explainable. Misalignment between image sets has both rigid and non-rigid components. Rigid registration followed by Pix2Pix (or Pix2Pix followed by rigid registration) cannot correct the some non-rigid misalignment and therefore is expected to have inferior performance compared to RegGAN.
>
> Q2: The results doesn't reflect actual utility in medical diagnosis.
>
> A2: We fully agree with you that performance metrics do not necessarily reflect clinical utility. It is a limitation of the current work and will be further investigated in our future studies. Effective and accurate evaluation of medical image-to-image translation is a rather complicated topic and requires careful consideration and significant amount of future work. Accurate medical image-to-image translation is a non-trivial task. It is not realistic to solve the entire problem in a single study. Like many other complicated projects, it requires many years of efforts to improve methodology and accumulate knowledge for the final breakthrough. Therefore, any steps that can bring us closer to the final goal are considered novel and meaningful. We also agree with you that the unpaired settings will exacerbate the situation you are worried about. Therefore, we pointed out in line 215 of the text that unpaired settings are not recommended in practice, just for exploring the stability and universality of the method.

---

### Decision · Program_Chairs · 2021-09-27

**Decision:**

Accept (Spotlight)

**Comment:**

This paper proposes a novel medical image-to-image generative model, called RegGAN, to combine an adversarially trained image generator with a registration module, which enforces the alignment between the generated image and noisy ground truth image. The paper illustrates the better performance of the proposed method on all major metrics for both paired and unpaired data and that RegGAN is more resistant to label noise and more stable in the training process. All reviewers agree that the results are convincing and that the work may bring insights to the image translation community.